# Enhancing portability of trans-ancestral polygenic risk scores through tissue-specific functional genomic data integration

Bradley Crone[1], Alan P. Boyle[1,2]*

**1** Department of Computational Medicine and Bioinformatics, University of Michigan, Ann Arbor, Michigan, United States of America, **2** Department of Human Genetics, University of Michigan, Ann Arbor, Michigan, United States of America

* apboyle@umich.edu

**Data Availability Statement:** Code is available in the TITR github repository: https://github.com/bcrone/TITR/.

**Funding:** This project was supported by NIH U24 HG009293 and NIH U01 HG011952 to APB. The

## Abstract

Portability of trans-ancestral polygenic risk scores is often confounded by differences in linkage disequilibrium and genetic architecture between ancestries. Recent literature has shown that prioritizing GWAS SNPs with functional genomic evidence over strong association signals can improve model portability. We leveraged three RegulomeDB-derived functional regulatory annotations—SURF, TURF, and TLand—to construct polygenic risk models across a set of quantitative and binary traits highlighting functional mutations tagged by trait-associated tissue annotations. Tissue-specific prioritization by TURF and TLand provide a significant improvement in model accuracy over standard polygenic risk score (PRS) models across all traits. We developed the Trans-ancestral Iterative Tissue Refinement (TITR) algorithm to construct PRS models that prioritize functional mutations across multiple trait-implicated tissues. TITR-constructed PRS models show increased predictive accuracy over single tissue prioritization. This indicates our TITR approach captures a more comprehensive view of regulatory systems across implicated tissues that contribute to variance in trait expression.

## Author summary

Polygenic risk score models leverage effect size estimates from ancestry-targeted GWAS to generate well-powered disease stratification models. When ancestry-targeted GWAS is unavailable for understudied populations, trans-ancestral PRS models may be implemented. However, transferring PRS models across ancestries results in limited predictive accuracy due to linkage differences between ancestries. Here we show that isolating GWAS variants with strong functional evidence identified from RegulomeDB-derived annotations in tissues enriched for trait heritability can improve portability of PRS models across distant ancestries. The motivation is mutations with evidence of regulatory impact are more likely to be shared between ancestries than genome-wide significant signals from ancestry-targeted GWAS. Further, we developed the novel TITR algorithm to aggregate functional GWAS mutations across multiple trait-implicated tissues to iteratively

funder did not play a role in the study design, data collection and analysis, decision to publish, or preparation of the manuscript.

**Competing interests:** The authors have declared that no competing interests exist.

construct PRSs. These models provide a more comprehensive view of functional GWAS mutations that influence variation in complex disease expression and can help improve portability of PRS models in under-represented populations.

## Introduction

Polygenic risk scores (PRS) are an effective statistical tool in stratifying complex disease risk. Risk stratification from PRS can help inform clinical decision making and treatments for high risk individuals, and avoid unnecessary treatments for individuals with low genetic risk [1]. The predictive power of PRS is a function of well-powered genome-wide association studies (GWASs) [2], where large clinical cohort studies produce stronger trait-variant associations, more accurate single nucleotide polymorphism (SNP) effect size estimates, and greatly reduces the chance of false positive disease associations [3]. However, large scale GWASs for most complex traits target individuals of European ancestry with over 79% of GWAS cohorts being of European origin despite representing only 16% of the global population [4]. Because of this disparity, PRS models developed for non-European individuals are under-powered and fail to stratify individual disease risk as effectively as European-centric models [5].

A handful of solutions are available to address this problem of under-powered PRS targeting non-European populations. The first is large-scale sequencing and genotyping initiatives targeting individuals from non-European ancestries. GWAS cohorts in which non-European populations are equitably represented should ensure robust and accurate associations and SNP effect sizes regardless of genetic background. Efforts are underway to achieve this goal of ancestrally-diverse biorepositories, including the All of Us Research Program [6], the Million Veterans Project [7], and BioBank Japan [8]. However, these and other initiatives are long-term solutions with significant cost barriers; meanwhile, precision medicine applications for under-represented populations require more-immediate solutions.

Trans-ancestral PRS models present an alternative solution that leverages large-scale European GWAS cohorts. Briefly, trans-ancestral PRS models take well-powered GWAS associations and effect size estimates from one population to predict and stratify disease risk in individuals of another population lacking a sufficiently-large GWAS cohort. The assumption is that mapping associations from a well-powered GWAS targeting one population to individuals in another population will capture a sufficient degree of SNP heritability across all ancestries [9]. However, mapping genome-wide significant associations alone results in poor transferability of scores due to differences in linkage disequilibrium (LD) patterns and genetic architecture between ancestral populations [10,11]. This bias is often driven by true causal signals in LD with tagging SNPs specific to one ancestry and not shared across other ancestries. Overcoming LD confounding remains a major obstacle in improving transferability of PRS models between ancestries.

By contrast, previous work has shown that prioritizing SNPs with strong functional genomic evidence improves PRS accuracy compared to traditional PRS computation, in which risk models are based on the SNPs with the strongest statistical associations. The general intuition is that selecting variants with functional evidence enriches the dataset for causal SNPs that likely contribute directly to function, thus circumventing the obfuscating effects of LD. An example method is AnnoPred [12], which assesses heritability enrichment across a set of functional annotations for a given GWAS trait. These annotation-specific enrichment estimates are then implemented in a Bayesian model as a prior distribution on GWAS SNP effect sizes. AnnoPred conferred greater predictive accuracy over PRS models generated from all GWAS

SNPs, genome-wide significant SNPs, SNPs prioritized by pruning and thresholding, and by Ldpred [13], which does not prioritize functional SNPs. This shows that introducing functional evidence of regulatory activity into SNP prioritization methods results in well-powered PRS models and better isolation of causal functional SNPs at GWAS loci. However, AnnoPred relies on well-matched training and target ancestry datasets, limiting its usefulness in cross-population PRS modeling. Recently, Amariuta et al. developed a technique for highlighting regulatory SNPs by leveraging cell-type-specific transcription factor (TF) binding profiles to construct risk models targeting trait-relevant functional SNPs [14]. This strategy provides significant accuracy improvement in trans-ancestral applications over the standard PRS approach. However, this is limited to evidence from a single cell-type-specific TF, whereas complex diseases are likely to have multiple regulatory features across many cell and tissue types, influencing variation in phenotypic expression. Annotations that combine multiple regulatory features can more accurately identify probable functional variants. Models like DeepSEA [15] and Sei [16] leverage sequence-based chromatin profiling data in deep-learning frameworks to produce allele-specific probabilities that effectively prioritize functional variants in GWAS data. These combined feature models significantly outperform single feature models in predicting chromatin features in non-coding genomic regions, and provide a framework for multi-omic integration in profiling important regulatory features across diverse tissue types. Additional approaches in combining heterogeneous genomic information like cooperative learning models [17] and Bayesian ensemble methods [18] can also boost signal and achieve higher predictive accuracy of functional genomic elements.

We hypothesize that prioritizing regulatory mutations on a tissue- or organ-wide scale within large European GWASs to construct a functionally-informed PRS can improve trans-ancestral portability of polygenic risk models when applied to under-represented non-European populations (Fig 1). Because of the differences in LD structures between ancestries, mutations with the highest probability of regulatory impact are more likely to be shared across ancestries than those with the strongest statistical association from the European GWAS. Here we introduce the TITR algorithm, a trans-ancestral PRS model that leverages SNP-level regulatory probabilistic scores from RegulomeDB. Three models–SURF [19], TURF [20] and Tland [21]—capture experimental evidence from the ENCODE project on either an organism-wide or tissue-specific scale and also feature allele-specific predictions from DeepSEA and Sei. We show that, by highlighting mutations with the greatest degree of organism-wide or tissue-specific regulatory evidence, we capture a significant proportion of SNP heritability and achieve greater predictive accuracy for trans-ancestral PRS compared to other published methods.

## Results

### Overview of non-coding regulatory mutations prioritization models

To prioritize functionally relevant GWAS mutations on a genome-wide basis, we leveraged the previously developed Score of Unified Regulatory Features (SURF) computational model. Briefly, SURF combines regulatory features from RegulomeDB [22,23], a database that annotates SNPs with known and predicted regulatory elements based on experimental data captured by the ENCODE project [24], and DeepSEA [15], a deep-learning framework designed to identify regulatory elements from large-scale chromatin profile data. These features are combined in a random forest model to produce probabilistic scores of regulatory function with SNP-level sensitivity [19]. In addition to the SURF model, we also employed tissue-specific regulatory scores from the Tissue-specific Unified Regulatory Features (TURF) model. TURF is an extension of the SURF model that isolates experimental evidence for a given tissue

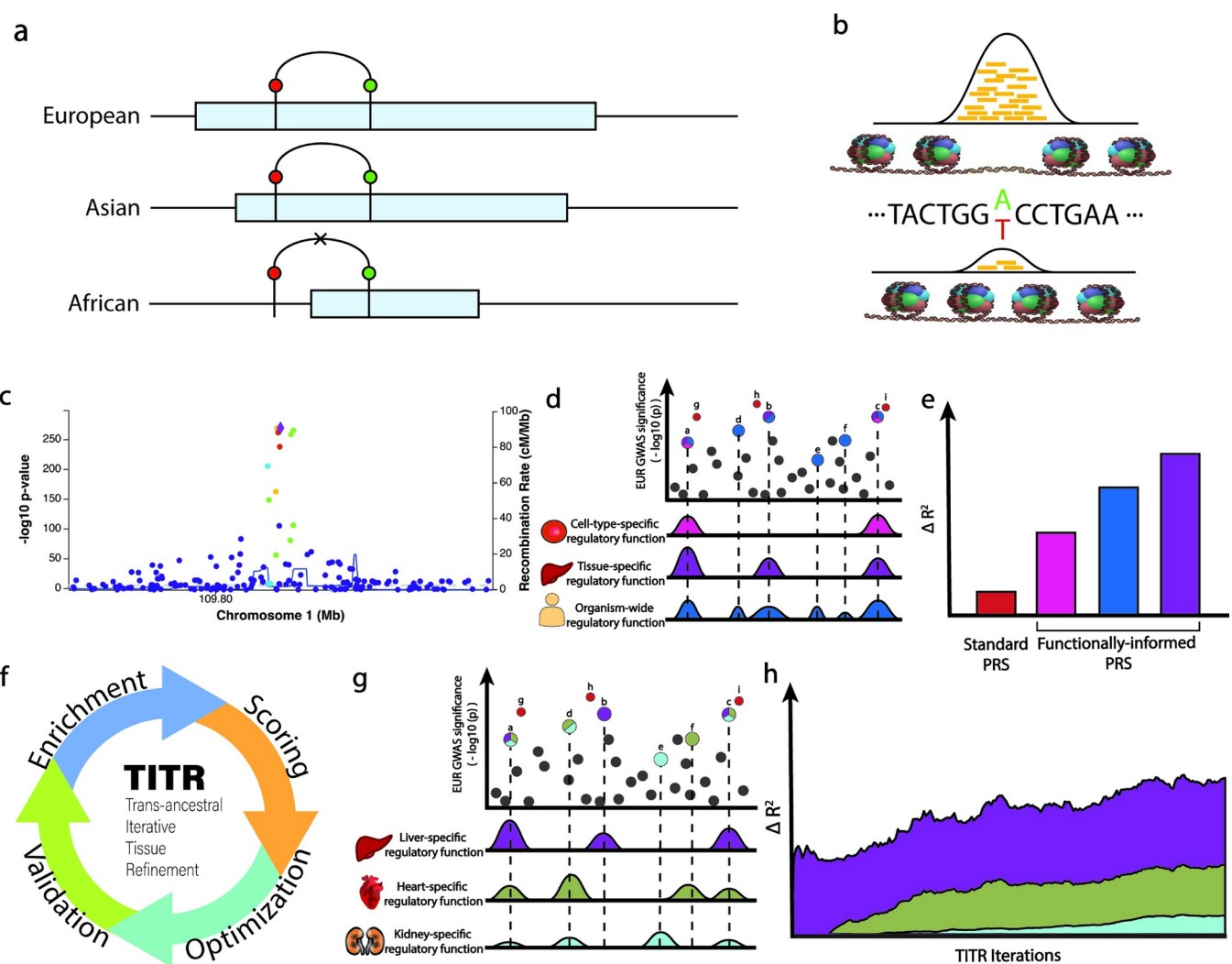

**Fig 1.** (a) Differences in LD structures between populations can lead to inaccurate accounting of disease loci in trans-ancestral risk modeling. Example given for a disease-associated locus with causal SNP (green) in strong LD with tagging/lead European GWAS SNP (red), where the associated LD blocks are represented as light-blue bars. Disease locus is accounted for in European and Asian structures, but not in African architecture, where the tagging SNP is not genetically linked to the causal SNP. (b) DNase-seq profiling can help identify allele-specific chromatin states where transcription factor binding may either be more readily accessible or completely ablated. (c) Determining causal variants at GWAS loci can be confounded by linkage with tagging variants (d) Prioritizing trait-relevant regulatory SNPs on either cell-type-specific, tissue-specific, or organism-wide functional evidence can help alleviate PRS confounding due to LD between SNPs, and (e) result in greater accuracy in disease risk modeling. (f) The TITR algorithm provides a more comprehensive picture of regulatory programs influencing variation in complex trait heritability by (g) combining multiple tiers of trait-relevant tissue-specific evidence (h) resulting in stronger predictions over a single implicated tissue model.

or organ present in ENCODE and feeds those features into a random forest model, weighted by organism-wide probabilities from the SURF model, to produce 51 distinct tissue-specific regulatory scores [20]. The goal of this model is to identify genomic regions with more significant regulatory effect within a specific tissue over organism-wide predictions. Finally, we also included the recently developed TLand model, which uses a stacked generalization model to learn RegulomeDB-derived features across all ENCODE hg38 experiments and predictions from Sei [16] to assign probabilities for regulatory variants on a cell-type and organ-specific level for 51 tissue-specific models [21].

## Partitioned SNP heritability by top percentiles of tissue-specific regulatory scores

To identify which set of regulatory SNPs in trait-implicated tissues explains the greatest proportion of heritability, we constructed percentile partitions of SNPs for all 51 ENCODE-defined tissues with TURF and TLand scores. These partitions encompass nearly 10 million SNPs from the 1000 Genomes Phase 3 European genotypes [25]. To ensure the percentile partition size was consistent across TURF scores, we performed quantile normalization on all TURF score distributions to make equivalent distributions to build the SNP partitions. This allows for equivalently sized percentile partitions across each tissue-specific score distribution. TLand scores are quantile normalized during generation. We estimated partitioned SNP heritability by leveraging the stratified LDSC method [26] for each TURF and TLand score independently, conditioning on a subset of baseline-LD model annotations [27], excluding annotations from datasets encompassed by ENCODE to avoid potential overfitting. We partitioned common SNP heritability (MAF > 5%) for 6 GWAS traits: 4 quantitative traits (height and BMI [28]; HDL and LDL cholesterol [29]) and 2 binary traits (coronary artery disease (CAD) [30] and type II diabetes (T2D) [31]). We selected the TURF or TLand model with the greatest, significant per-standardized-annotation effect size (denoted $\tau^*$), which is defined as the proportional change in per-SNP heritability attributed to a 1 standard deviation change in annotation value [27]. The motivation for using this statistic as our selection criteria is to highlight the functional annotation with the greatest impact on SNP heritability differences as a function of annotation variance. We compared these heritability estimates against published results from IMPACT [14].

Our results show the TURF and TLand models with the greatest $\tau^*$ estimates align well with implicated tissues for several traits (S1 Table and S1 Note). Some TURF examples include adipose tissue for HDL cholesterol [25,32,33], liver tissue for LDL cholesterol [34,35], and pancreas for type 2 diabetes [36]. Some highlighted TLand examples include endocrine gland for HDL cholesterol [37,38] and large intestine for LDL cholesterol [39,40]. The lead TURF model for height was mouth and may have resulted from misclassifications in experimental assays present in the ENCODE database. For example, several fibroblast assays, implicated in connective tissues, are tagged as mouth tissue.

To demonstrate that the lead TURF models capture a large proportion of trait heritability within the upper quantiles of the scoring distribution, we assessed the proportion of heritability captured by SNPs tagged across multiple score quantiles of the lead TURF and TLand model. We partitioned heritability for all GWAS traits by the top 1/5/10/20/50% of the lead TURF model (selected by $\tau^*$ criteria described above) (S2 Table). On average, 6.7% (SE = 0.0137) and 6.6% (SE = 0.0128) of observed scale heritability is captured by SNPs tagged within the Top 1% of the lead TURF and TLand model distributions, respectively. This increases to 25.4% (TURF) and 28.2% (TLand) captured by the Top 5%, and 41.0% (TURF) and 43.2% (TLand) by the Top 10% in the scoring distribution. Overall, both TURF and TLand lead models capture a large proportion of trait heritability with the top percentiles of the scoring distribution, concluding that these tissue-specific multi-feature predictions effectively highlight functional GWAS variants.

In summary, we identified the top TURF and TLand tissue-specific models for a set of GWAS traits that substantially align with trait-implicated tissues or organs, and also produce modest per-standardized-annotation effect size estimates. These top TURF and TLand models also capture a large proportion of overall trait heritability within the top percentiles of their respective scoring distributions. This demonstrates that SNPs with a greater degree of functional evidence account for a larger proportion of trait heritability than SNPs without strong regulatory evidence.

## Variant prioritization by TURF modeling improves trans-ancestral PRS portability

As noted before, PRS models provide insight into an individual's relative genetic risk of disease compared to a large spectrum of other genetic risk profiles. However, accurate PRS models require GWASs with large ancestry-specific population samples, which is problematic when constructing risk models targeting underrepresented non-European ancestries. To address the need for greater predictive accuracy in trans-ancestral PRS applications, we implemented pruning and thresholding (P+T) models restricted to different sets of GWAS variants with strong trait associations, mutations in genomic coding regions, and functionally-relevant variants in impacted tissues and cell types. We developed a baseline linear model composed of the covariates age, sex, principal components 1–10 from PCA analysis, and a coding PRS model that isolates known common coding mutations (MAF > 5%) as an additional covariate. The motivation behind incorporating this coding score as a covariate in our baseline model is to separate out the trait heritability of regulatory regions from those of coding variants. This is required to explicitly demonstrate the added benefit of regulatory functional prioritization in the generation of our PRS models. Since our regulatory function predictions do not predict coding impact of variants and we know that coding variants drive a significant amount of heritability, failing to regress on coding signal would not allow for comparison to the standard model PRS. Our baseline model (labeled Standard in Fig 2) takes the standard P+T approach, which clumps and prunes GWAS SNPs based on strength of association from the GWAS

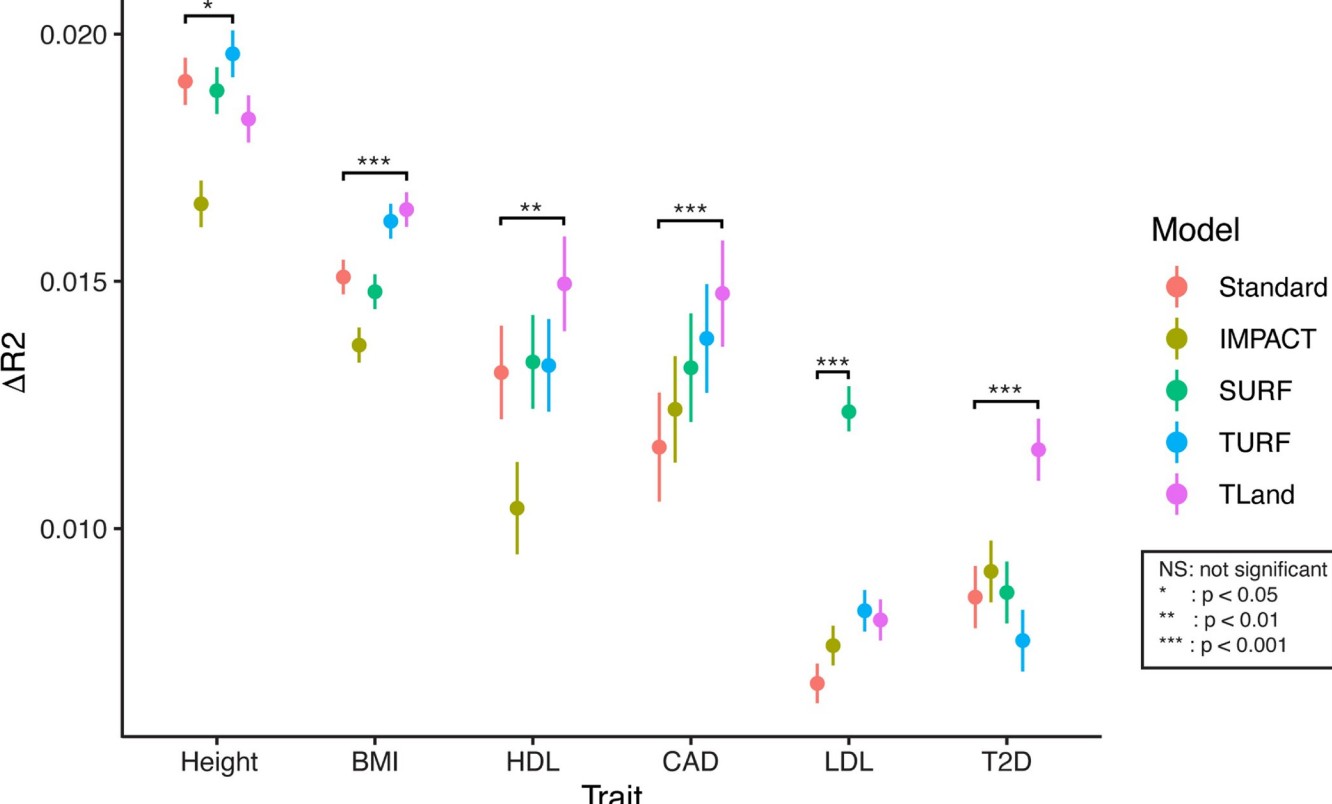

**Fig 2. Change in R² accuracy of EUR-AFR trans-ancestral PRS models by functional SNP prioritization model.** Results for 6 phenotypes (4 quantitative—BMI, height, HDL and LDL cholesterol; 2 binary—coronary artery disease (CAD), type 2 diabetes (T2D)). Error bars indicate 95% confidence intervals calculated by 1000 bootstraps. Significance is indicated as lead functional model performance against standard model performance.

summary statistics. We then constructed PRS models for both lead TURF and TLand tissue functional annotations as determined by S-LDSC functional enrichments, and the lead IMPACT TF annotation from published data for each GWAS trait. We partitioned scores by multiple thresholds in the respective annotation distributions: top 1%, top 5%, top 10%, top 20%, and top 50% of GWAS SNPs for each annotation. We also included a model that prioritizes SNPs with general functional activity from the previous SURF model, using the same score partitions as above (top 1/5/10/20/50%). We clumped SNPs based on the 1000 Genomes Phase 3 European reference panel (n = 489) [25], and validated scores for individuals of African ancestry present in the UK Biobank (n = 7,324) [41] (Trait sample summaries are detailed in S3 Table). For the functional annotation models, we selected the partition with the largest change in adjusted $R^2$ accuracy over the baseline linear model as the representative model for the lead annotation.

We observed for all 6 traits tested that the lead functionally-informed model significantly outperformed the standard PRS model in terms of $R^2$ accuracy gained over the null model of covariates (Fig 2). Of these, 4 traits—BMI, HDL, CAD, and T2D - saw the greatest improvement with the lead TLand model, 1 trait—height—saw the greatest improvement with the lead TURF model, and 1 trait—LDL—saw the greatest improvement with the lead SURF model. For the 5 traits where either TLand or TURF models provide the greatest accuracy gains, this indicates that a greater burden for polygenic risk can be derived from a single-tissue-specific regulatory model by TLand or TURF than from a general regulatory model provided by SURF, or cell-type-specific model by IMPACT. This is contrasted by the results observed for LDL where we infer that regulatory information from a single TLand or TURF model isn't sufficient compared to the general SURF regulatory model. Overall, we confirm that prioritization of functional regulatory GWAS SNPs on a tissue- or organism-wide scale in transancestral PRS models provides a significant improvement in phenotypic prediction accuracy over both cell-type-specific TF prioritization and standard PRS approaches.

## Iterative construction of a multiple-tissue functional PRS model improves accuracy over a single-tissue prioritization model

Complex polygenic diseases can affect multiple tissues and organs throughout the human body. Diseases like coronary artery disease and type 2 diabetes are largely localized to a single organ or tissue type and most of the trait heritability can likely be explained by our previously described single-tissue functional prioritization method. However, single-tissue prioritization is likely insufficient for traits like BMI and height that involve multiple organs and tissues. This is because single-tissue methods only highlight functional mutations within the most significant tissue while ignoring regulatory effects in other relevant tissues. To address this, we developed the Trans-ancestral Iterative Tissue Refinement (TITR) algorithm to iteratively construct a multiple-tissue functional PRS model that captures relevant regulatory mutations across multiple trait-implicated tissues (Fig 3).

Briefly, the TITR algorithm takes European GWAS summary statistics and annotates SNPs with tissue-specific functional probability scores and corresponding percentile bins of the scoring distribution for all 51 TURF models and 51 TLand models. The algorithm then estimates the partitioned trait heritability for each tissue model and selects those models significantly enriched for trait heritability. Polygenic risk scores are calculated for the candidate tissues and optimized in European samples, selecting the scoring model that produces the greatest significant $\Delta R^2$ over the previous model iteration. The algorithm terminates if no tissue model is significantly enriched for heritability, or if no score provides a significant addition of information to the risk model (See Methods for detailed algorithm description).

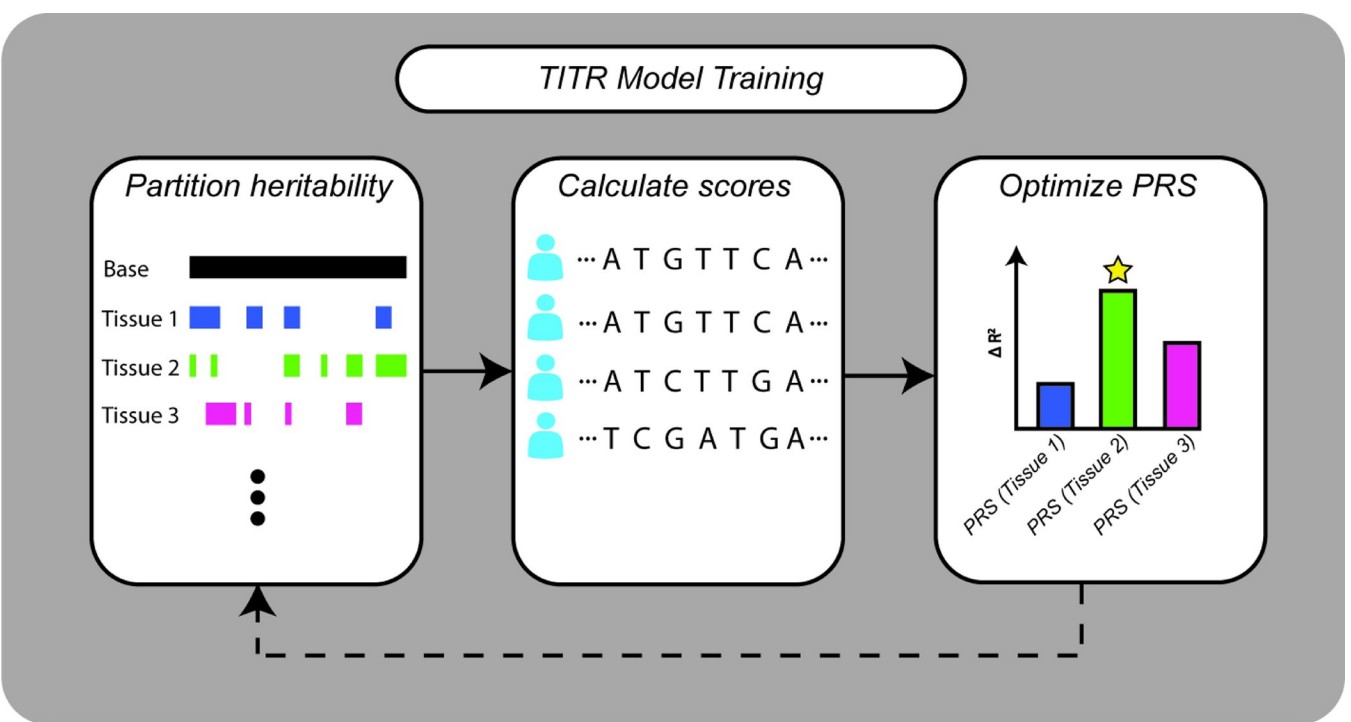

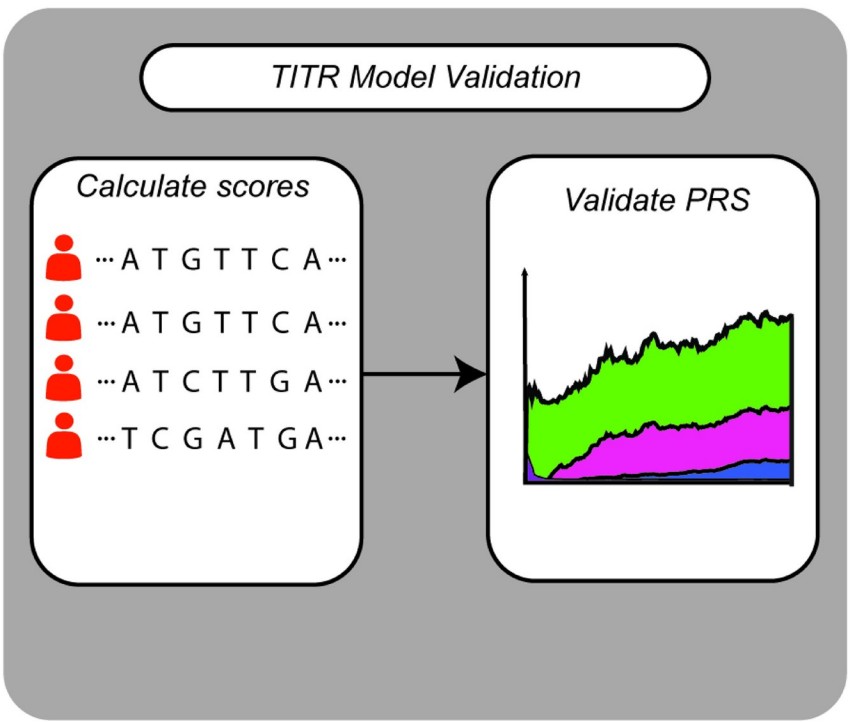

**Fig 3. Workflow of the TITR algorithm.** Training of the TITR model involves: 1) Partitioning trait heritability by functional annotation; 2) Calculating scores within training samples; and 3) selecting the optimal PRS model on each iteration. Validation of the TITR model involves: 1) Calculating scores within validation samples; and 2) calculating the PRS accuracy increase and tissue annotation contributions over TITR iterations.

To validate our TITR-constructed European-optimized PRS model in non-European population samples, we implemented a trans-ancestral PRS model to utilize the optimized TITR model SNP set and constructed risk scores targeting disease risk in individuals of African ancestry. For our analysis, we selected a cohort of self-reported African ancestry individuals (n = 7,324) from the UK Biobank. We constructed risk models for 4 quantitative traits: BMI, height, HDL and LDL cholesterol; and 2 binary traits: CAD and T2D. As with the European optimization model, we assessed the $\Delta R^2$ over a null model consisting of the covariates age, sex, first 10 principal components, and the previously-optimized coding scoring model.

We assessed the performance of the TITR-optimized PRS model against the performance of the single tissue prioritization model, comparing the change in adjusted $R^2$ accuracy ($\Delta R^2$) for the top 1% of model SNPs, and the lead scoring partition of SNPs for both TURF and TLand lead tissue models. We report the TURF model performance across the first 200 iterations of TITR for BMI, height, and LDL cholesterol in Fig 4. Results for TURF and TLand annotations for the remaining 3 traits—HDL cholesterol, coronary artery disease, and type 2 diabetes—are reported in S1 and S2 Figs (legend for tissue labels in S5 Fig). We observed for all traits that the TITR model performance exceeded the accuracy gains for the top 1% of lead TURF and TLand prioritized SNPs. This is consistent with our expectation that the TITR algorithm preferentially selects trait-relevant SNP partitions with strong regulatory evidence over non-relevant mutations with weak regulatory evidence. To further support this expectation, we calculated the overlap of GWAS SNPs captured by both the top 1% of TURF or TLand single-tissue models and TITR-optimized models (S3 and S4 Figs). For the 3 highlighted traits BMI, height, and LDL cholesterol, we observed that TITR models capture 54.9–77% of the top 1% of TURF SNPs and 34–75% of the top 1% TLand SNPs, indicating a considerable overlap between trait-relevant SNPs and high probability of functional impact on regulatory systems. However, when we compare TITR model accuracy against the lead SNP partitions defined in the single-tissue approach for TURF and TLand, 8.8–32.2% of lead TURF partition SNPs and 9.8–42.6% of lead TLand partition SNPs overlap with TITR model SNPs. The limited SNP overlap and enhanced predictive accuracy of TITR PRS models over the lead single-tissue PRS model indicates that the TITR algorithm better isolates functional mutations impacting regulatory systems across multiple trait-implicated tissues than prioritizing mutations within a single tissue.

### Assessing functionally-informed PRS models in other non-European populations demonstrates similar trans-ancestral portability improvements

While modeling trans-ancestral PRSs from European to African populations demonstrates the power of functional prioritization between distant ancestries, modeling between less distant ancestries should also demonstrate similar power improvements over standard PRS modeling approaches. We evaluated both single-tissue and TITR-optimized PRS models in a set of UKBB SAS samples (n = 7,515). RegulomeDB-derived functional PRS models conferred the greatest $\Delta R^2$ over null (BMI: 0.039 (TURF); height: 0.0448 (TURF); HDL: 0.0155 (TURF); LDL: 0.00779 (TLand); CAD: 0.0135 (TURF); T2D: 0.0175 (SURF)). TITR-optimized models outperformed the top 1% TURF model for 4 of 6 traits tested (BMI, height, LDL, T2D) and outperformed the top 1% TLand models for 3 of 5 traits tested (BMI, height, T2D). Overall, tissue-specific functional prioritization in SAS validation samples conferred larger accuracy improvements over baseline null models and standard P+T as in AFR validation samples.

### Discussion

Here we have shown that leveraging functional genomic data when constructing polygenic risk models improves trans-ancestral portability of PRS over selecting those GWAS variants

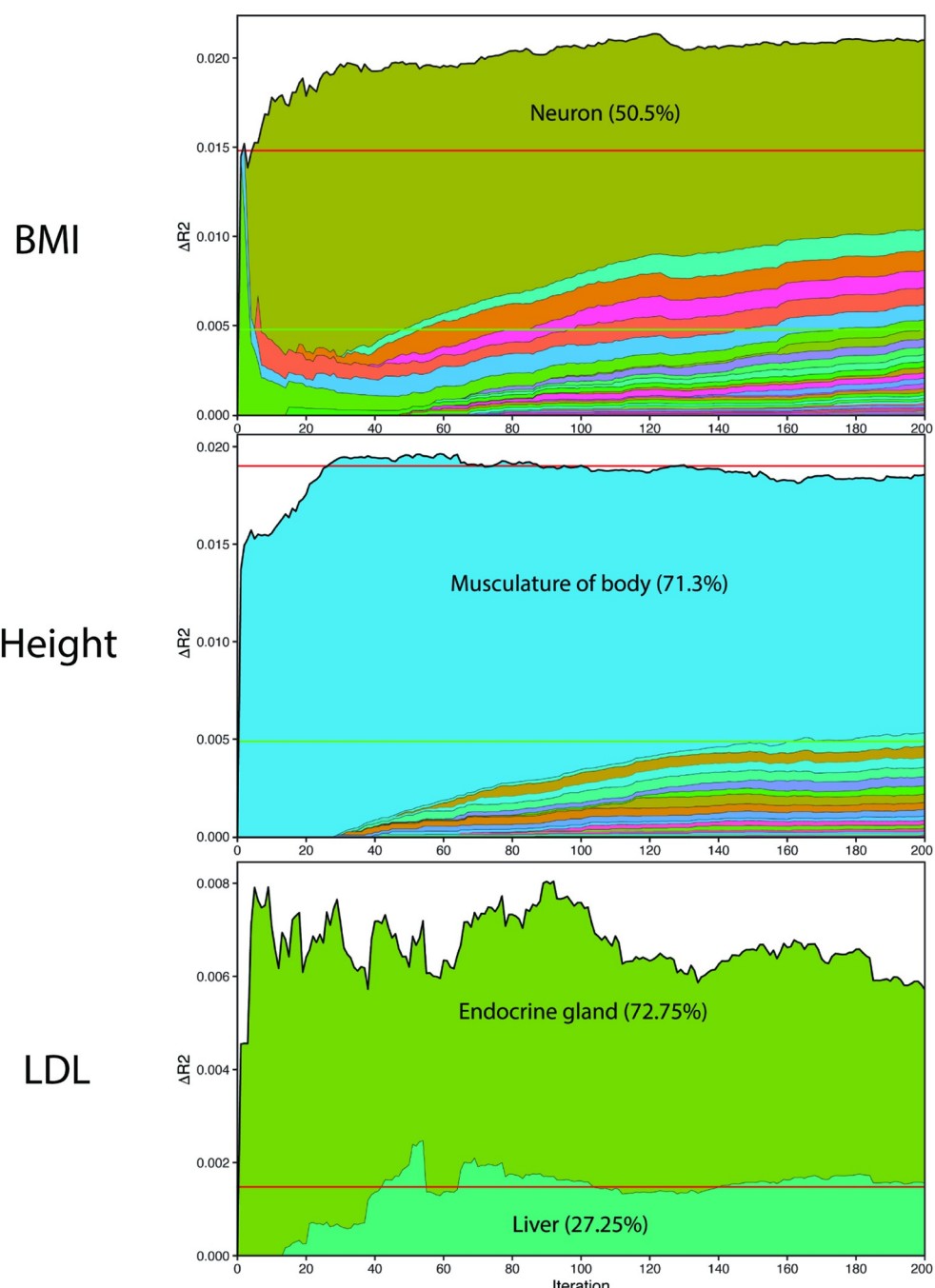

**Fig 4. Adjusted R$^2$ accuracy of EUR-to-AFR trans-ancestral PRS model up to the first 200 iterations of the TITR optimization algorithm for TURF functional model.** Results are shown for 3 example traits—BMI, height, and LDL cholesterol. Dashed black horizontal lines indicate performance by the single TURF model targeting the top 1% SNPs in scoring distribution. Solid black horizontal lines indicate performance by the single TURF model targeting the lead partition SNPs in scoring distribution. The most represented tissue models and SNP proportion in the TITR-optimized PRS model indicated by labels.

with the strongest statistical associations. By prioritizing functionally enriched tissue-specific TURF and TLand regulatory annotations identified by stratified LDSC partitioned heritability enrichments, we provide a modest yet significant improvement in predictive accuracy of

European-derived PRS models when applied to individuals of African ancestry. These data support our hypothesis that isolating functional mutations specific to trait-implicated organs and tissues encapsulated by RegulomeDB-derived annotation models can provide a more comprehensive picture of disease architecture across ancestries than population-specific GWAS loci alone.

With the construction of ensemble risk models by the TITR algorithm, we also demonstrate that polygenic disease risk can be more accurately explained across multiple regulatory pathways within trait-implicated tissues and organs. Two example traits where regulatory information from multiple tissues improved risk prediction accuracy in our African validation cohort are BMI and LDL cholesterol, both prioritizing GWAS SNPs based on TURF functional annotation scores. For BMI, the lead S-LDSC enriched TURF tissue model was brain. The single tissue PRS approach selected SNPs in the top 50% of the TURF brain annotation, providing a 31.4% increase in predictive accuracy over the null model. Under the TITR algorithm, a majority of functional prioritized SNPs were selected from bipolar neuron tissue (50.5%, labeled "ear" by the ENCODE project) compared to a small fraction from general brain tissue (1.01%) resulting in a 45.3% accuracy improvement over the null model. For LDL cholesterol, the lead S-LDSC enriched TURF tissue model was liver. The single tissue PRS approach selected SNPs in the top 20% of the TURF live annotation, providing a 93.1% predictive accuracy increase over the null model. Under the TITR algorithm, the model incorporates functional information from endocrine gland tissue SNPs (72.75%) as well as liver tissue (27.25%) resulting in a greater than 5-fold increase in predictive accuracy over the null model. These examples indicate that TITR is better at isolating trait-specific functional mutations as a function of (1) selecting the tissue model partition with the greatest change in $R^2$ accuracy across all significantly-enriched tissues and (2) finer granularity in SNP partitioning as opposed to the 1/5/10/20/50% partitioning under the single tissue model. We show here that by incorporating evidence of regulatory mechanisms affected by a given polygenic disease across several affected tissues, we can further improve our ability to stratify disease risk for individuals of non-European descent by refining GWAS findings from European cohorts.

We note a few limitations to our study. First, we leverage European-derived effect size estimates, which may be biased towards linkage patterns found in European populations. A future direction of this approach is to incorporate effect size re-scaling similar to that implemented by LDpred-funct [42], which could provide more accurate estimates of allelic effects and better downstream modeling of trans-ancestral disease risk. Second, the TITR algorithm provides more powerful predictions over the single-tissue approach only where trait heritability is significantly enriched across multiple tissues. This is evidenced by the performance of HDL cholesterol and coronary artery disease under the TURF functional model (S1 Fig), where only adipose tissue and thyroid gland are significantly enriched for partitioned trait heritability across all iterations of TITR, respectively. In these cases, risk models generated from the single-tissue approach provide the best performance in trans-ancestral applications and should be selected over the TITR multiple tissue approach.

Finally, we note RegulomeDB-derived functional PRS models perform poorly compared to standard P+T approaches when tested within the same ancestry (Europeans). This highlights a critical distinction in LD structure differences between populations. LD is greater in Europeans due to genetic bottlenecks from out-of-Africa migrations, resulting in shorter recombination time compared to African populations who have not experienced similar bottlenecks [43]. Because of stronger LD in European populations, tagging SNPs in European-derived GWAS are likely in LD with true causal SNPs, thus potentially inflating effect size estimates of tagging SNPs due to the "winner's curse" phenomenon [44]. Thus, isolating causal signal with functional prioritization for within-ancestry European likely attenuates predictive accuracy as

models do not incorporate inflated tagging signal. Conversely, since LD is weaker in African populations, European-derived tagging SNPs are less likely to be in LD with true causal SNPs. Therefore, functional prioritization in African-targeted PRS models could capture causal signals otherwise unaccounted for under standard P+T conditions. Overall, these results indicate standard P+T approaches select genome-wide significant variants specific to European populations and confer greater accuracy for within-population predictions, whereas functionally-informed models select variants with probable regulatory impacts shared between populations and provide stronger predictive power in trans-ancestral models.

Overall, we have demonstrated the predictive power of highlighting functional mutations on a tissue- or organ-wide scale in trans-ancestral polygenic risk modeling, and described a framework to iteratively build risk models that target functional mutations across multiple implicated tissues for polygenic traits. This serves as a step in better isolating and identifying shared causal mutations in regulatory programs across ancestries and developing more accurate risk models for under-represented populations.

## Methods

### Genome-wide association data

We obtained publicly-available summary statistics for 8 GWAS traits—4 quantitative, 4 binary —constructed from cohorts of European individuals. We selected these traits for the large European training samples (average $n_{quantitative} \approx 1M$, average $n_{binary} \approx 287K$). GWAS summary statistics were collected from multiple sources: the GIANT Consortium for height and BMI; the Global Lipids Genetics Consortium for HDL and LDL cholesterol; the CARDIoGRAMplusC4D Consortium for CAD; and the DIAGRAM Consortium for T2D. For S-LDSC analysis, summary statistics were formatted to contain the following information for each SNP: rsID, reference allele (A1), alternative allele (A2), effective GWAS sample size per SNP (N), and the chi-square statistic derived from the GWAS betas (Z). For the polygenic risk score calculation, summary statistics were formatted to contain the following information for each SNP: rsID (SNP), alternative allele (A1), SNP effect size estimate (BETA), and GWAS association signal (P). Example GWAS summary stats with headers are available in the GitHub repository.

### Functional genomic annotations

We leveraged 4 functional genomic annotation datasets to partition GWAS variants. The first is IMPACT, which models cell-type-specific transcription factor binding profiles to identify probable regulatory features genome-wide. Further details on the IMPACT model are reported in Amariuta et al. 2020 [14]. The second is SURF, which predicts regulatory features organism-wide by modeling transcription factor binding sites leveraging ENCODE experimental assays and deep learning predictions from DeepSEA. Further details on the SURF model are reported in Dong et al. 2019 [19]. The third is TURF, an extension of SURF, to generate tissue- and organ-specific predictions of regulatory features genome-wide. Further details on the TURF model are reported in Dong et al. 2022 [20]. The fourth is TLand, which constructs a stacked generalization model to learn RegulomeDB-derived features to predict regulatory variants at a cell-specific level or organ-specific level. Further details on the TLand model are reported in Zhao et al. 2023 [21].

### UKB genotypes

For PRS analysis, we utilized phenotype and genotype data from the UK Biobank. UK Biobank is a large-scale biomedical database and research resource containing genetic, lifestyle and

health information from half a million UK participants. We identified 2 sets of genotype samples with associated phenotype and demographic information to train and validate both single-tissue and multiple-tissue TITR models. For model training, we randomly selected 40,000 individuals who self-reported as being of white British European ancestry. For model validation, we selected 7,324 individuals who self-reported as being of African ancestry. For each of the traits tested in this study, we identified subsets of each sample set with either quantitative phenotype measurements, or constructed case/control subsets for binary traits, maintaining a 4:1 control to case ratio.

## Partitioned heritability by functional category

We used stratified LDSC v1.0.1 to partition SNP heritability for our set of 8 GWAS traits. We used a custom set of 23 baseline-LD annotations that do not overlap with ENCODE datasets and incorporated one of 51 tissue-specific TURF annotations into the model. We calculated per-annotation standardized effect sizes ($\tau^*$) for each partitioned model as the proportional change in per-SNP heritability associated with a 1 standard deviation increase in the value of the annotation. For single tissue model construction, we selected the TURF annotation with the greatest significant, non-zero $\tau^*$ value after Bonferroni correction ($\tau^* > 0, p(\tau^*) < \frac{0.05}{51}$) as the lead TURF annotation for the given trait. For TITR model construction, we selected all TURF annotations with significant non-zero $\tau^*$ values after Bonferroni correction as candidate tissues on each iteration.

## TITR algorithm overview

The TITR model starts by taking European GWAS summary-level statistics and annotates each SNP with corresponding functional probabilities and percentile bins of the scoring distributions from all 51 TURF tissue-specific models. For our analysis, we generated bins with 0.1% resolution resulting in 1000 partitions for each of the 51 TURF models. The iterative construction and optimization in European training samples of our TITR PRS was achieved as described in Table 1.

## LD block sampling method

To protect against over-sampling SNPs within the same LD block, we implemented a custom SNP selection algorithm that prioritizes functionally-relevant mutations over strongly-associated variants in the GWAS. On the 1st iteration of the TITR algorithm, we build an LD structure for the GWAS trait with the PLINK clumping tool, using the 1000 Genomes European genotype set as a reference panel. For a given TURF tissue partition, we select representatives from LD blocks containing SNPs from that partition, preferentially selecting index variants, followed by the most GWAS-significant clumped SNP in the block. This process selects at most 1 SNP per LD block. For subsequent iterations, we repeat this procedure and merge the newly-selected SNPs with the current set of TITR model SNPs to then calculate scores.

## Polygenic risk score calculation

We calculated polygenic risk scores using the PLINK 2.0 linear scoring tool for candidate SNP sets for each significant TURF tissue partition. For each partition, we generated scores across a spectrum of 370 p-value thresholds. We invoked the 'no-mean-imputation' option to ignore missing or unnamed alleles in genotypes, rather than allowing PLINK to add proportional weight by imputed allele frequency. We calculated scores across a training set of European UKB samples ($n$ = 40,000) and testing set of African UKB samples ($n$ = 7,324).

**Table 1. TITR Algorithm Overview. The workflow of the TITR algorithm is shown describing each step, the tools used, specific conditions, and the output.**

| Algorithm Step | Description | Tools/Datasets | Conditions | Output |
|---|---|---|---|---|
| Partition $h^2$ | Estimate $h^2$ and calculate $\tau^*$ for each functional annotation | Stratified LDSC GWAS summary statistics Functional annotation models | $\tau^* > 0$ p < 0.05/51 (Bonferroni correction) | List of prioritized candidate functional annotation models |
| Calculate scores | Calculates allelic scores for lead partitions of prioritized functional annotation models in training samples | PLINK GWAS summary statistics UKB EUR training samples | Clump $r^2 = 0.2$ Tests 370 p-value thresholds $p \in [0.00001, 1]$ | PLINK scoring profiles for UKB EUR training samples |
| Optimize PRS | Constructs linear regression models for each candidate tissue scores, selects model with greatest $\Delta R^2$ over current model | Custom Python tools PLINK scoring profiles | 1st iteration: passes ANOVA test for nested linear models Subsequent iterations: passes Cox test for non-linear models | Master model log (tissue, partition, $\Delta R^2$, SNP count, etc.) PRS model SNP list Regression coefficients for training samples |
| Validate PRS | Generates allelic scores for AFR validation samples Constructs linear regression models across all iterations | PLINK Custom Python tools GWAS summary statistics UKB AFR validation samples | None—final model has passed all conditions in optimization | Validation model log (tissue, partition, $\Delta R^2$, SNP count, etc.) Regression coefficients for validation samples |

## PRS optimization

We optimized our PRS model on each iteration of the TITR algorithm. To do this, we constructed a null linear model regressing sample phenotype values against a set of covariates: age, sex, the first 10 principal components, and an independently-optimized coding-region-specific PRS. We then tested all scores for each tissue and set of p-value thresholds against the null model. For the first iteration, we select candidate scores that add significant information to the model by conducting an ANOVA test for nested linear models. For subsequent iterations, we select candidate scores that add significant information to the model by conducting a J test for non-nested linear models. From the candidate scores, we select the lead score as the score that provides the greatest change in predictive accuracy ($\Delta R^2$) over the previous model.

## PRS validation

We validated our European-optimized PRS model across iterations of TITR. To do this, we calculate scores for each TITR iteration within a set of African ancestry samples from the UK Biobank. Then, we constructed a null linear model regressing sample phenotype values against a set of covariates: age, sex, the first 10 principal components, and an independently-optimized coding-region-specific PRS. For each iteration, we add the PRS model as an additional regressor in the linear model. We then assess the change in predictive accuracy ($\Delta R^2$) of each iteration's scoring model.

## Supporting information

**S1 Note. Increased coverage of variants scales $\tau$.**
(DOCX)

**S1 Fig. Adjusted R2 accuracy of EUR-to-AFR trans-ancestral PRS model for HDL cholesterol, coronary artery disease, and type 2 diabetes of the TITR optimization algorithm for**

**TURF functional model.**
(TIF)

**S2 Fig. Adjusted R2 accuracy of EUR-to-AFR trans-ancestral PRS model for BMI, height, HDL cholesterol, LDL cholesterol, and type 2 diabetes of the TITR optimization algorithm for TLand functional model.**
(TIF)

**S3 Fig. Proportion of SNPs overlapping between TURF single tissue model (top 1/5/10/20/50% thresholds) and TITR PRS model for BMI, height, HDL cholesterol, LDL cholesterol, coronary artery disease, and type 2 diabetes.**
(TIF)

**S4 Fig. Proportion of SNPs overlapping between TLand single tissue model (top 1/5/10/20/50% thresholds) and TITR PRS model for BMI, height, HDL cholesterol, LDL cholesterol, and type 2 diabetes.**
(TIF)

**S5 Fig. Legend for tissue labels in TITR results–Figs 3, S1 and S2.**
(TIF)

**S1 Table. Lead tissue annotations by largest significant tau\* S-LDSC prioritization for TURF, TLand, and IMPACT functional models.**
(XLSX)

**S2 Table. SNP heritability captured by top 1/5/10/20/50% TURF and TLand annotations.**
(XLSX)

**S3 Table. UK Biobank African sample descriptions.**
(XLSX)

**S4 Table. Significance of differences in mean dR2 for single tissue PRS models between standard, IMPACT, SURF, TURF, and TLand PRS models.**
(XLSX)

**S5 Table. Overlap of index and LD clumped SNPs between single tissue model SNPs and TITR prioritized SNPs.**
(XLSX)

**S6 Table. Null model R2 values for each trait by population.**
(XLSX)

## Acknowledgments

This research has been conducted using the UK Biobank Resource under Application Number 24460. We thank Shengcheng Dong, Nanxiang Zhao, and Rintsen Sherpa for their help in generating SURF, TURF, and T-Land annotations. We also want to thank all members of the Boyle Lab for their support and constructive feedback.

## Author Contributions

**Conceptualization:** Bradley Crone, Alan P. Boyle.

**Formal analysis:** Bradley Crone.

**Funding acquisition:** Alan P. Boyle.

**Investigation:** Bradley Crone.

**Methodology:** Bradley Crone, Alan P. Boyle.

**Project administration:** Alan P. Boyle.

**Software:** Bradley Crone.

**Supervision:** Alan P. Boyle.

**Visualization:** Bradley Crone.

**Writing – original draft:** Bradley Crone, Alan P. Boyle.

**Writing – review & editing:** Bradley Crone, Alan P. Boyle.

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
