## [Decision Letter · Decision Letter 0]

20 Mar 2024

Dear Dr Boyle,

Thank you very much for submitting your Research Article entitled 'Enhancing Portability of Trans-Ancestral Polygenic Risk Scores through Tissue-Specific Functional Genomic Data Integration' to PLOS Genetics.

The manuscript was fully evaluated at the editorial level and by independent peer reviewers. The reviewers appreciated the attention to an important problem, but raised some substantial concerns about the current manuscript. Based on the reviews, we will not be able to accept this version of the manuscript, but we would be willing to review a much-revised version. We cannot, of course, promise publication at that time.

If you decide to revise the manuscript for further consideration at PLOS Genetics, please aim to resubmit within the next 60 days, unless it will take extra time to address the concerns of the reviewers, in which case we would appreciate an expected resubmission date by email to plosgenetics@plos.org.

We are sorry that we cannot be more positive about your manuscript at this stage. Please do not hesitate to contact us if you have any concerns or questions.

Yours sincerely,

Yun Li

Academic Editor

PLOS Genetics

Michael Epstein

Section Editor

PLOS Genetics

Reviewer's Responses to Questions

**Comments to the Authors:**

Reviewer #1: Review uploaded as an attachment.

Reviewer #2: I think a flowchart or visualization for the PRS construction and TITR workflow would be very helpful. I would also like to see a comparison method that incorporates functional annotations (not just standard P+T) to assess the improvement transferability of your method. LDPred-funct may be easy since it uses S-LDSC, which you have already calculated, or alternatively PANPRS.

Reviewer #3: This study presents a new method, TITR, which leverages tissue-specific functional annotation information to increase PRS accuracy, especially to increase the accuracy of EUR GWAS based GWAS in non-EUR target data. The authors argue that TITR can improve the PRS calculated using Pruning and P-value thresholding with annotation information from existing annotations by iteratively constructing a multiple-tissue functional PRS model.

Even though this is a topic of interest in the field, I would like to suggest some comments and questions

TITR is supposed to increase the PRS accuracy by choosing PRS more likely to be causal instead of just in LD the the causal SNPs. If this is true, TITR can also increase PRS within the EUR population. As a supplementary test, the author can also test the optimized PRS in EUR samples.

To give a more comprehensive evaluation, the optimized PRS should also be tested in other non-EUR populations. For example, it is very convenient to test the PRS in SAS samples in UKBB.

Similarly, it will be very helpful if the authors could test the method with more traits. The existing results focus on metabolic-related traits. The result of more diverse traits (e.g. cancer-related, auto-immune disease-related, neural system/ psychiatric-related) would be much appreciated.

The authors need to give some clarification/ explanation about why their null model is “a set of covariates: age, sex, the first 10 principal components, and an independently-optimized coding-region-specific PRS”, since this practice is not very common in the field at least to my best knowledge. Even though the R^2 increase on top of the null model R^2 shows whether the increase exists or not, it is necessary to provide the baseline value to get the idea of the relative increase, which could be of more interest when evaluating the method. Besides that, I would suggest the authors also give a more commonly used set of results where the null model contains only the covariates and the PRS optimized by TITR is compared with PRS constructed using the naive P+T method. Since the main highlights of this method is the additional PRS accuracy increase gained from the iterative construction of the PRS model, I hope the author can also provide the R^2 over the iteration calculated using this more commonly used way.

Since it has become increasingly common to use SNP effect size adjusted by Bayesian methods, for example LDpred2 and PRS-CS, and an existing method, X-Wing (Miao et al, 2023), has already combined Bayesian method with annotation, I wonder if the author would like to consider the possibility of incorporating the TITR with Bayesian adjusted SNP effect size. Should this be beyond the scope of this project, the author should at least include the likelihood and their suggestion on doing so in the discussion

Minor comments:

The author should give more detailed information of the summary statistical data (download link if available, release date, and the sample size for each GWAS data) since the many consortiums listed in the method session released multiple batches of data. Besides that, the exact R^2 values or delta R^2 used for generating the plot should be given in the supplementary.

**Have all data underlying the figures and results presented in the manuscript been provided?**

Reviewer #1: Yes

Reviewer #2: Yes

Reviewer #3: **No: **The author should give more detailed information of the summary statistical data (download link or release date, and the sample size for each GWAS data) since the consortiums listed in the method session released multiple batches of data. Besides that, the exact R^2 values or delta R^2 used for generating the plot should be given in the supplementary.

PLOS authors have the option to publish the peer review history of their article (what does this mean?). If published, this will include your full peer review and any attached files.

Reviewer #1: No

Reviewer #2: No

Reviewer #3: No

---

## [Decision Letter · Decision Letter 1]

27 Jun 2024

Dear Dr Boyle,

We are pleased to inform you that your manuscript entitled "Enhancing Portability of Trans-Ancestral Polygenic Risk Scores through Tissue-Specific Functional Genomic Data Integration" has been editorially accepted for publication in PLOS Genetics. Congratulations!

Yours sincerely,

Yun Li

Academic Editor

PLOS Genetics

Michael Epstein

Section Editor

PLOS Genetics

Comments from the reviewers (if applicable):

Reviewer's Responses to Questions

**Comments to the Authors:**

Reviewer #1: I appreciate and thank the authors for addressing all the reviewer comments.

Reviewer #2: This paper provides a new algorithm TITR, which leverages tissue-specific annotations to increase the portability of PRS across different ancestries. The study is well conducted and addresses an important issue. The authors have included figures and tables that have clarified each step of the algorithm. Additionally, the authors have provided prediction in non AFR ancestries and provided baseline estimates.

**Have all data underlying the figures and results presented in the manuscript been provided?**

Reviewer #1: Yes

Reviewer #2: Yes

PLOS authors have the option to publish the peer review history of their article (what does this mean?). If published, this will include your full peer review and any attached files.

Reviewer #1: No

Reviewer #2: No

**Data Deposition**

http://datadryad.org/submit?journalID=pgenetics&manu=PGENETICS-D-24-00261R1

**Press Queries**

---

## [Editor Report · Acceptance letter]

2 Aug 2024

PGENETICS-D-24-00261R1 

Enhancing Portability of Trans-Ancestral Polygenic Risk Scores through Tissue-Specific Functional Genomic Data Integration 

Dear Dr Boyle, 

We are pleased to inform you that your manuscript entitled "Enhancing Portability of Trans-Ancestral Polygenic Risk Scores through Tissue-Specific Functional Genomic Data Integration" has been formally accepted for publication in PLOS Genetics! Your manuscript is now with our production department and you will be notified of the publication date in due course.

With kind regards,

Anita Estes

PLOS Genetics

On behalf of:
